# Analysis of P4 and XDP for IoT Programmability in 6G and Beyond

**David Carrascal** [1,†] , **Elisa Rojas** [1,*,†] , **Joaquin Alvarez-Horcajo** [1,†] , **Diego Lopez-Pajares** [2] **and Isaías Martínez-Yelmo** [1]

[1] Departamento de Automática, Escuela Politécnica Superior, University of Alcala, 28801 Alcalá de Henares, Spain; david.carrascal@uah.es (D.C.); j.alvarez@uah.es (J.A.-H.); isaias.martinezy@uah.es (I.M.-Y.)

[2] Departamento de Ingeniería de Sistemas Telemáticos, E.T.S de Ingenieros de Telecomunicación, Technical University of Madrid, 28040 Madrid, Spain; diego.lopezp@upm.es

* Correspondence: elisa.rojas@uah.es

† These authors contributed equally to this work.

**Abstract:** Recently, two technologies have emerged to provide advanced programmability in Software-Defined Networking (SDN) environments, namely P4 and XDP. At the same time, the Internet of Things (IoT) represents a pillar of future 6G networks, which will be also sustained by SDN. In this regard, there is a need to analyze the suitability of P4 and XDP for IoT. In this article, we aim to compare both technologies to help future research efforts in the field. For this purpose, we evaluate both technologies by implementing diverse use cases, assessing their performance and providing a quick qualitative overview. All tests and design scenarios are publicly available in GitHub to guarantee replication and serve as initial steps for researchers that want to initiate in the field. Results illustrate that currently XDP is the best option for constrained IoT devices, showing lower latency times, half the CPU usage, and reduced memory in comparison with P4. However, development of P4 programs is more straightforward and the amount of code lines is more similar regardless of the scenario. Additionally, P4 has a lot of potential in IoT if a special effort is made to improve the most common software target, BMv2.

**Keywords:** IoT; programmability; P4; XDP; SDN; 5G; 6G; edge computing

## 1. Introduction

After an initial and successful deployment of the first commercial 5G networks, there is growing hype around the innovation outcomes expected for the next generation of wireless communications technologies supporting cellular data network, i.e., 6G [1,2]. Even considering that 6G is already framed in diverse white papers (such as the Network 2030 envisaged by the International Telecommunication Union (ITU) [3]), the specific requirements of this technology are still nebulous.

Notwithstanding, a clear consensus exist around the paramount importance of Internet of Things (IoT) [4,5] in the design of future 6G networks. Some predictions expect around 34 billion connected devices by 2024, of which two third parts will be basically IoT-based, with a growth of 10% per year (5G Alliance for Connected Industries and Automation (5GACIA): https://www.5g-acia.org/) [6]. Some other works emphasize on the importance of next generation IoT connectivity to boost edge-computing together with Artificial Intelligence (AI) [7], and the design of scalable and energy-efficient machine type communications [8], which are demands of the future digital society that currently 5G cannot accomplish [9].

One of the core reasons why 5G is not capable of achieving a thorough integration of IoT is that its enabling technologies were not designed to meet the heterogeneous requirements of

edge devices [10]. In particular, Software-Defined Networking (SDN) offers high flexibility and programmability, which are key features for 5G in comparison with traditional networks. However, it remains a challenge to apply this technology in IoT environments, often constrained in energy, composed of wireless unreliable links and, thus, unable to assume the control overhead imposed by SDN [11].

As a conclusion, there is an urge to find a solution that overcomes the current limitations of SDN to guarantee a comprehensive integration of the IoT ecosystem into the future deployment of 6G. For this purpose, in this article we delve into two different technologies, namely Programming Protocol-independent Packet Processors (P4) [12] and Express Data Path (XDP) [13] that, together with SDN, could potentially foster the seamless integration of IoT into next-generation 6G networks. Our intention is to provide a quick overview of their capabilities, including current opportunities and challenges, as a reference for forthcoming research efforts by including a deep analysis of each technology that contains:

- The development of a set of network scenarios, publicly available for replication, to test the features of XDP and P4.
- The evaluation and comparison of XDP and P4 in terms of programmability, performance as well as a qualitative assessment of both technologies.
- A discussion on the suitability of both XDP and P4 in IoT networks, focusing on Low-Power and Lossy Networks (LLNs), and defining some future work in the field.

To the best of our knowledge, no previous work exists considering all three aspects at the same time. Therefore, the main contribution of the paper, in comparison to the state-of-the-art, is to collect in a document the differences between two of the newest technologies used in SDN environments and to assess their applicability in the challenge of IoT in 6G networks, providing a clear baseline for future research.

The remainder of this manuscript is structured as follows: In Section 2, we summarize the related work. Afterwards, Sections 3 and 4 are devoted to explain the applied methodology and results obtained for the analysis, respectively. Section 5 discusses the previous sections, trying to provide a hint on future research lines, while Section 6 concludes the paper.

## 2. Related Work

One of the main achievements in 5G networks is the definition of the Multi-access Edge Computing (MEC), which is implemented by leveraging SDN and constitutes the backbone for the practical implementation of edge [14,15] and fog computing [16,17]. These new computing architectural approaches enable an enhanced interaction of the core network with the edge nodes, including IoT. However, standard 5G architectures do not encompass IoT devices as part of their softwarized network [18], but mainly as entities to be served by it.

Although a significant interest exists in applying SDN-like programmability in IoT [19], diverse open challenges remain such as reducing the mobility control overhead, or increasing fault tolerance [20,21]. Minh et al. [22] emphasize on three pillars to overcome these limitations, namely openness, scalability, and programmability. Foster et al. [23] present an inspirational white paper that also advocates for the need for an open-source deep programmability across the stack, both vertically (control and data plane) and horizontally (end to end). Whereas Omnes et al. [24] envision how NFV and SDN can help facing the upcoming challenges in IoT. Nevertheless, none of these works actually design an SDN framework including constrained, yet programmable, IoT devices.

Alternatively, some recent proposals indirectly tackle the integration of IoT in SDN environments. For example, the Hybrid Domain Discovery Protocol (HDDP) [25] permits the discovery of non-SDN devices (including IoT sensors) by installing a simplified agent. However, additional features are required for a fully fledged management of IoT. Another example is Whisper [26], which indirectly controls IoT networks by injecting artificial but standard-compliant messages. Nonetheless, its current

approach is only compatible with networks based on the IPv6 Routing Protocol for LLNs (RPL) and does not support other protocols, such as Lightweight On-demand Ad hoc Distance-vector routing protocol-next generation (LOADng). Therefore, these proposals still require some improvements.

Diverse technologies exist strictly focused on network data plane programmability. P4 [12] is a declarative language for programming network devices, particularly focused on process and forward actions. By definition, it aims to be decoupled from the hardware it should control. Network devices that support P4 are defined as *targets*, and the standard target is the Behavioral-Model (BMv2) (https://github.com/p4lang/behavioral-model). At the same time, P4Runtime (https://github.com/p4lang/p4runtime) might be used to communicate any target programmed in P4 with an SDN controller. *µ*P4 [27] provides a higher-level abstraction for P4 to facilitate the writing of portable, modular and composable programs. The P4 language has also been applied to extend the well-know software switch Open vSwitch (OVS), resulting in P4rt-OVS [28]. An example of IoT programmable data planes leveraging P4 is IoTP [29]. In addition, XDP [13], usually together with extended Berkeley Packet Filter (eBPF) [30], offers high-performance in-kernel packet processing by adding a hook in the Linux kernel network stack and providing a set of predefined actions. hXDP [31] is a solution to execute programs written in eBPF on FPGAs, usually constrained in resources, which is the case of IoT networks. P4C-XDP [32] benefits from combining both technologies: P4 and XDP, as it compiles P4 into code for an XDP driver. Finally, another technology worth mentioning is Data Plane Development Kit (DPDK) [33], a set of libraries designed to accelerate packet processing workloads. However, as it is mainly focused on boosted performance for high-capacity devices, it falls out of the scope of our analysis.

Although P4 and XDP provide frameworks to flexibly design network data planes, no specific effort has been made on IoT devices and most works are focused on non-constrained data planes. At the same time, diverse authors are recently calling on the need to bring these features right to the very edge of networks. As a conclusion, there is a need to further extend network data plane programmability to constrained IoT environments. In particular, extensive research effort is required towards the integration of SDN in LLNs, comprised of IoT devices with limited memory and energy. In this regard, the novelty of our work is to provide an initial framework to foster research efforts in this area.

## 3. Materials and Methods

In order to examine the capabilities of P4 and XDP for their application in LLNs, which constitutes one of the pillars of 6G networks, we defined three main types of analysis:

1. Basic functionality evaluation
2. Performance assessment
3. Qualitative review

The first analysis is devoted to provide a very quick overview of how basic functionality can be deployed in network devices by using either P4 or XDP. In the end, as we are targeting constrained IoT devices (also known as *motes*), it is necessary to have a glimpse of the unitary functions that could be programmed and be executed individually, without depending on additional resources. The second analysis attempts to illustrate the actual performance achieved by each technology, which might be critical in certain scenarios. In the end, the main objective is to have a trade-off between functionality and performance. Finally, the qualitative review aims to serve as a simplified guide for developers that are new in the field.

### 3.1. Definition of Basic Use Cases

As seen in Section 2, P4 and XDP technologies have many strengths, and both allow the definition of the datapath of a network device (Note that a *datapath* could be defined as a collection of functional units, such as arithmetic logic units or multipliers, that perform data processing

operations. In the specific context of communication networks, a datapath defines packet processing and forwarding). While P4 proposes a language, intended to be hardware-agnostic, to describe the datapath, which increases the adaptability of network protocols to a huge range of hardware platforms, XDP focuses on the packet processing in the interface, which limits the adaptability of the protocols but enables a reactive-mode operation that increases performance and deletes active consumption of resources.

Taking into account these premises, we defined a set of basic functionalities that a hypothetical IoT device would have to implement. These functionalities are going to be gathered in use cases, which were developed with both technologies (P4 and XDP) to conclude which one is simpler and more efficient in terms of programmability, a key aspect towards the integration of the IoT devices in SDN environments. The selected five use cases were the following:

1. Drop: To test the most basic function, which is dropping any received traffic.
2. Pass: Together with drop, this function simply passes/delegates the traffic to any secondary process/listener (if any) without affecting the data plane.
3. Echo server: To check whether the device can actively reply to traffic.
4. L3 forwarding: A slightly more advanced function that acts on the packet based on a routing table, i.e., the network device is able to understand layer-3 (L3) information.
5. Broadcast: This action requires the ability not only to act on the packet, but to duplicate it and act on their copies.

In addition, we considered the two main types of physical interfaces, i.e., wireless and wired environments. This is particularly relevant to explore the benefits of both technologies in gateway IoT devices that are partially connected to the SDN core in a wired way and, at the same time, through a wireless interface with the rest of IoT devices.

In wired scenarios, the evaluations of the XDP programs were carried out on a Linux platform in which network topologies were initially generated using Network Namespaces [34], to isolate each network node, and Veth [35], to emulate the links between different nodes. An example of this is depicted in Figure 1a. As for the platform chosen to evaluate the P4 programs, as depicted in Figure 1b, Mininet was used, which is a tool to emulate SDN network topologies. Mininet [36] was selected because the p4lang team developed an interface (https://github.com/p4lang/behavioral-model/tree/master/mininet) of the BMv2 with that tool, so it is possible to incorporate the reference soft-switch to test P4 code (BMv2), as a type of another node in Mininet. Mininet could also be used for the XDP scenarios as, in the end, it leverages both Network Namespaces and Veth to deploy the diverse network scenarios.

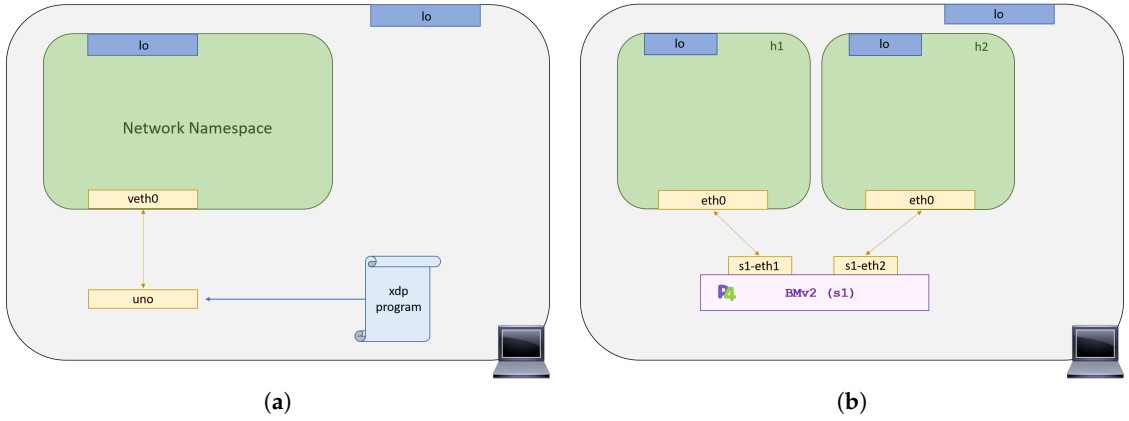

(**a**)                                                                                                   (**b**)

**Figure 1.** Example of the developed wired scenarios: (**a**) XDP wired scenario (**left**); (**b**) P4 wired scenario (**right**).

In the case of the wireless scenarios, the platform used for the evaluation of the different use cases, both XDP and P4, is the same: the Mininet-WiFi [37] emulator. This emulator emerged as an extension of Mininet that added support for wireless networks in SDN environments. It is important to point out that this platform did not contemplate any kind of node supporting BMv2, so it was necessary to previously develop an integration of BMv2 in Mininet-WiFi. An example of these deployments is shown in Figure 2.

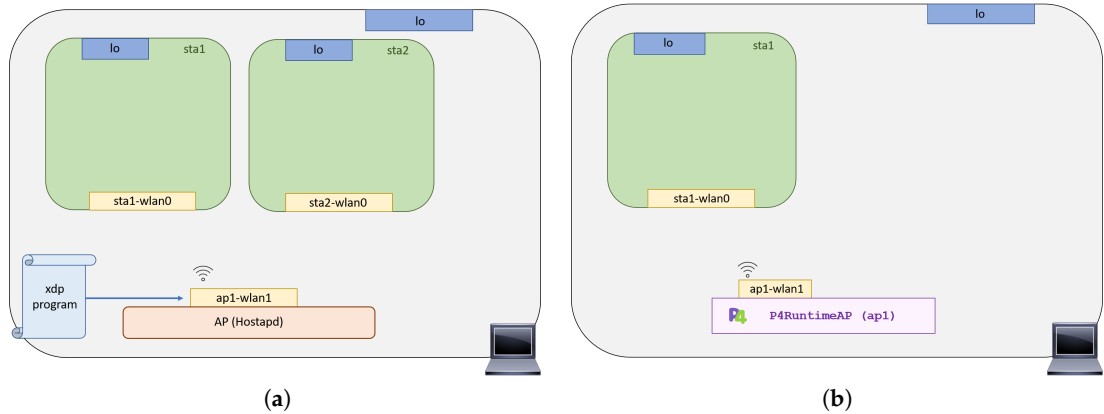

| (**a**) | (**b**) |

**Figure 2.** Example of the developed wireless scenarios: (**a**) XDP wireless scenario (**left**); (**b**) P4 wireless scenario (**right**).

Please note that the reason to select Linux-based software was on the basis of using the same system for all performed tests. At the same time, leveraging Linux is the easiest way to implement functionality in IoT devices. Therefore, results should consider this hypothesis and might differ if the target changes.

Contributions and Code Repositories

The complete implementation of all use cases, both XDP and P4, in both scenarios (wired and wireless) is available as open source in GitHub [38]. In this repository any reader can find the source code, as well as scripts for the installation of dependencies, startup and cleanup of the different use-case scenarios, and step-by-step guides to replicate the results presented here.

Additionally, as previously mentioned, it was necessary to develop an interface of the BMv2 software-switch with Mininet-WiFi in order to evaluate the use cases in wireless scenarios. This implementation is available as well in GitHub [39] and it was submitted to the official Mininet-WiFi repository as a pull-request. We encourage any eager reader to test the different implementations and provide any feedback, which will be highly appreciated.

*3.2. Performance Testbench*

The purpose of these experiments is to evaluate the performance gap between both technologies in terms of three parameters: latency, memory, and CPU consumption. These parameters provide a high-level overview of the two technologies to assess its energy performance in constrained environments. Figure 3, describes the testbench architecture used, which consists of two virtual machines connected via an internal network within VirtualBox, in which all machines can interact with each other through the adapter supplied by VirtualBox, with a capacity of 1 Gbps.

In this scenario, we defined two nodes called Traffic Generator and Receiver (TGR) and System under Test (SUT). The TGR machine generates and receives packets, hence stressing the SUT machine to obtain different latency metrics, packet losses, and maximum throughput for different percentages of CPU consumption. The characteristics of the SUT machine (see Table 1) were carefully chosen to be able to draw conclusions about the performance of both technologies in devices with limited resources.

All experiments were repeated 10 times. The results and tools used for data collection and processing can be found in GitHub [40] so that anyone can replicate the tests carried out.

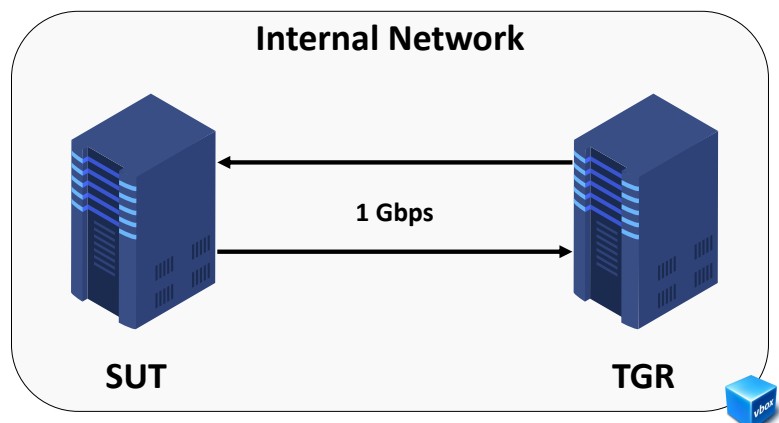

**Figure 3.** Testbed architecture

**Table 1.** SUT Hardware characteristics.

| Type | Characteristics |
|------|-----------------|
| CPU | Intel(R) Core(TM) i5-7500 CPU (1 Core) at 3.40 GHz |
| RAM | 4096 MB |
| NICs | Intel PRO/1000 MT 1Gb |
| OS | Ubuntu 18.04 |
| Kernel | Linux 5.4.0-42-generic |

### 3.3. Qualitative Review Method

Finally, for the qualitative study, five different points were studied: (1) documentation quality, (2) independence from other software, (3) current integration in hardware, (4) migration and updates, and (5) potential of integration in SDN environments. The first parameter, documentation quality, indicates how good is the documentation provided by the organizations behind the different technologies. Secondly, independence from other software represents the amount of software needed by each technology to operate, the fewer dependencies the better. In the case of hardware integration, we examined how compatible each technology is with the current hardware. The migration and update parameter indicates how easy is for a developer to install and update the same code on different targets. Finally, potential integration in SDN environments evaluates the potential to incorporate each technology in SDN devices today. These parameters were evaluated in a relative scale from 0 (most negative) to 10 (most positive) by our research team, taking into consideration the work performed for this article, but also previous developments.

Additionally, we collected several statistics from Google Scholar and GitHub to provide a trend overview of both technologies. The information from Google Scholar was directly obtained with their available filters, using keywords and publication year, whereas to collect the data from repositories in GitHub for both technologies, we leveraged the official GitHub rest Application Programming Interface (API). In order to work with this rest API, several scripts [40] were developed to automate data extraction and subsequent processing. The total number of repositories for both technologies was obtained by making a search request of the existing repositories in GitHub, in P4 they were filtered by language, since GitHub recognizes the P4 language. As for XDP, a more restrictive repository search request had to be made, including only repositories with C language syntax and with unequivocal XDP program parameters. In order to obtain the number of active repositories, different temporal parameters returned by the rest API were processed, which indicated the last-update dates of the previously filtered repositories.

## 4. Results

In the following subsections, we show and evaluate the results obtained from the three types of analysis performed for both P4 and XDP, following the methods and materials previously detailed in Section 3.

### 4.1. Basic Functionality Evaluation

The first analysis is devoted to illustrate the results obtained from the five use cases, defined to serve as tests of unitary functionality of P4 and XDP. After the five use cases are presented and evaluated, a final subsection summarized and remarks the main conclusions obtained in this analysis.

#### 4.1.1. Case I—Drop

In the first use case, we explored whether it was possible to drop packets with both technologies. The proposed scenario consisted of three nodes, two of which wanted to communicate through the third one, and the third one would be in charge of discarding the packets that passed through it. It was possible to see, independently of the environment, that it was feasible to develop such functionality with either XDP or P4, either by using return codes in XDP, specifically `XDP_DROP`, or by using primitives as it is the case of P4, specifically `mark_to_drop()`. Therefore, in this case, no significant difference between the two technologies could be obtained.

#### 4.1.2. Case II—Pass

In the second use case, the aim was to check whether it is possible to let packets go through (to any secondary process, if available) without affecting the data plane programmed with each technology. The scenario is completely analogous to that of the previous use case. There is a series of nodes where two of them want to communicate through a third one, but in this case, the packets will not be discarded, they will be let through, to allow communication between the nodes.

Using XDP, it could be seen that, although it is normally conceived as a mechanism to make a by-pass to the network stack of the Linux Kernel (i.e., a complete redefinition of it), in many cases it will be useful to work together to achieve the desired functionality. This action of delegating the packet will be carried out by using the `XDP_PASS` return code.

As for P4, the main conclusion was that deploying this exact functionality was not feasible, since there is no equivalent in P4 of the `XDP_PASS` return code. This was one of the big differences between both technologies indeed. With XDP the packet can always be passed to the Kernel to be processed by it, but in P4, the whole datapath must be defined exclusively, so there is no one to delegate the packet to (unless you specifically defined it with a table entry). This is a big plus for XDP, since, ultimately, you will always have the entire Linux Kernel processing stack for those packets that you do not know how to handle.

#### 4.1.3. Case III—Echo server

The main objective of the third use case was to evaluate the capability that both technologies have to filter and parse packets based on their headers. To this purpose, we developed a server that answers all the `ECHO-Request` messages that arrive to it. The scenario was similar to the previous use cases, from one node we tried to ping another node in the topology, and an intermediate node, running the Echo server, answered those pings before they reach the final destination.

In both technologies it was possible to implement the desired functionality, regardless of the scenario. The methodology to follow in both was similar, first the packets were filtered, by using the data structures of the different protocols to be processed in XDP, and in P4, by using data structures called *headers* to define the headers of the protocols to be processed and of parsers (status machines).

In the case of an `ECHO-Request` message, the link layer and network layer addresses were swapped, the ICMP header fields were modified to generate the response, and finally, the packets were forwarded

back through the same interface at which they arrived. In XDP, it was possible to carry out the action by using the return codes, in this case XDP_TX. In P4, the metadata associated with the packet was used to specify its outgoing port. So in this use case, it was seen that it was viable to filter packets and associate a logic to that filtering with both technologies.

### 4.1.4. Case IV—Layer 3 Forwarding

In this case of use, we explored how straightforward it is to forward packets from one interface to another on a device. The scenario proposed for this use case consists of a series of nodes that want to communicate through a third one, but the third one must manage to carry out the forwarding correctly for the communication to be established.

Different ways of doing forwarding in XDP were explored using Berkeley Packet Filter (Berkeley Packet Filter (BPF)) helpers, from the simplest and most static one in which the forwarding information is hardcoded in the bytecode itself, to the most complex one based on working collaboratively with the Kernel to obtain the forwarding information as shown in Figure 4. P4 uses metadata to carry out the forwarding of the packets, while the control plane via P4Runtime defines the matching tables in the BMv2 to define by which ports the packets should egress.

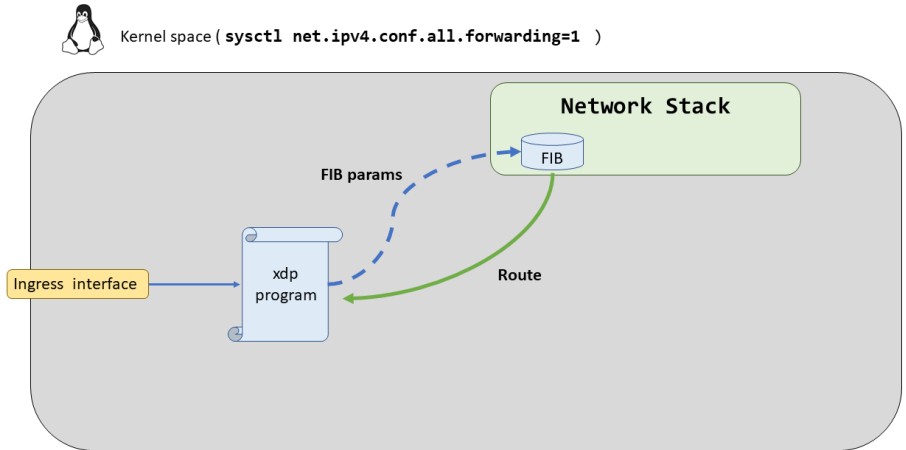

**Figure 4.** XDP Forwarding Auto.

### 4.1.5. Case V—Broadcast

Finally, with the fifth use case we aimed to address how the flooding (or broadcast) of a hypothetical packet can be achieved, i.e., if it is possible to clone packets with these technologies or, alternatively, create packets from scratch. The scenario proposed for both technologies in both scenarios was to broadcast the ARP-Request packets from an Address Resolution Protocol (ARP) resolution.

In both scenarios, wired and wireless, the P4 technology was able to execute the broadcast by using the so-called *multicast groups*, a high-level interface where the packet replicas are defined. In the case of XDP, in wired environments it was not possible to use the associated BPF helper for the packet cloning since it required that the packet was already governed by a sk_buff type structure. Therefore, it was necessary to consider a hybrid solution between the XDP program hooked in the interface, and another BPF program hooked in the Traffic control (TC). However, for XDP in wireless environment, it was already feasible to carry out the functionality since it was not necessary to clone the packet due to the wireless environment properties. It was only necessary to modify the headers of the packet as broadcast and forward it through the same interface at which it arrived.

### 4.1.6. Use Cases Feasibility

Table 2 summarized the results obtained from all the previous use cases. According to this summary, XDP was not able to perform a native broadcast in wired environments [41]. Nonetheless,

this limitation was studied, and a solution was proposed, making use of an additional BPF program in the TC to be able to clone the packets. On the other hand, if this functionality is required in a wireless environment, there would be no such limitation since it would not be necessary to clone the packets. Generally speaking, changing the destination address of the packet from layer two to broadcast, and transmitting it to the medium, will be sufficient to achieve the broadcast. On the contrary, in P4 technology, unlike XDP, it has a high level interface to define broadcast-multicast functionalities called *multicast groups*.

**Table 2.** Summary of the feasibility of the proposed use cases

| Use Cases | XDP-Wired | XDP-Wireless | P4-Wired | P4-Wireless |
|---|---|---|---|---|
| Case I—Drop | ✓ | ✓ | ✓ | ✓ |
| Case II—Pass | ✓ | ✓ | ✗ | ✗ |
| Case III—Echo server | ✓ | ✓ | ✓ | ✓ |
| Case IV—Layer 3 forwarding | ✓ | ✓ | ✓ | ✓ |
| Case V—Broadcast | ✗ | ✓ | ✓ | ✓ |

Looking back to Table 2, we want to emphasize that P4 technology must be able to manage all possible packets that come to it, so the missing functionality is to delegate packet actions, as by definition an action is required to do so. On the other hand, with XDP packets can always be delegated to the Kernel so that it takes care of the processing. This point is very positive for XDP, since simple and repetitive actions can be defined in an XDP program and delegate the rest of functionalities to the Kernel, achieving a much higher performance.

Finally, we would like to mention the difference between the two technologies in the control interface. In P4, the control interface is represented by the tables, which define their structure from the P4 program itself, but will usually be populated via P4Runtime by an external controller. On the other hand, in XDP we could say that the equivalent to the tables would be the BPF maps. These maps, of key-value type, are defined by the program that is hooked up to the Kernel, and they are generally completed by user space programs that access it through file descriptors. Therefore, currently P4 technology has a control interface more prepared for an immediate integration in SDN networks, leaving XDP in a weak position in this regard.

*4.2. Performance Assessment*

In this section the performance of both technologies is presented in a quantitative way, considering the three parameters defined in Section 3.

4.2.1. Latency

The methodology followed to test the latency of each technology was to load the use case III (described in Section 4.1.3), where with P4 and XDP an Echo server was developed. In particular, we implemented a packet generator in the TGR with an increasing amount of packets per second. These packets were directed to the SUT machine, which processed them, modified each request to generate the associated response, and forwarded them back through the same interface. These tests have been repeated a total of 10 times, so that we could evaluate on average the maximum and minimum values of latency associated with each technology.

As it can be seen in Figure 5, the results are quite promising in favour of XDP, with a maximum latency value even lower than the minimum latency value for P4. It can be concluded that the cost of processing packets in the Kernel makes the processing time of packets in XDP almost worthless. On the opposite, in P4, as it is using BMv2, which is ultimately a user space process, it is limited by the whole data plane implemented in the Linux Kernel network stack, besides being stopped every time there is an Interrupt Request (IRQ).

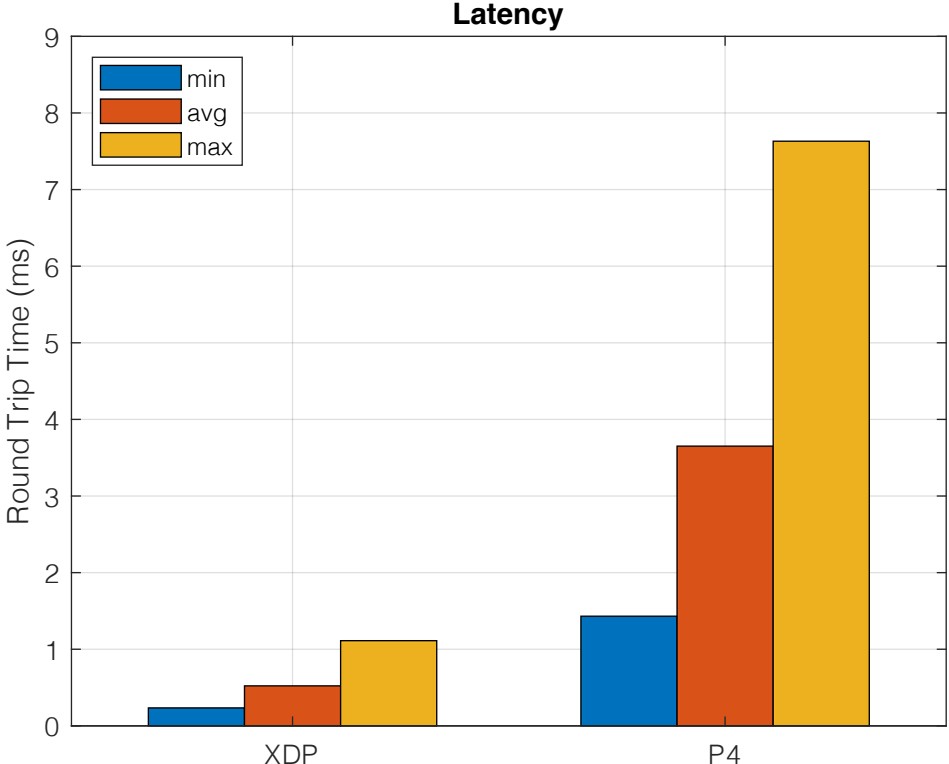

**Figure 5.** Latency of XDP vs. P4.

### 4.2.2. Memory Consumption

The main challenge in this second test was to obtain the memory consumption from XDP. It was easier for P4 because, as the BMv2 is a process running in user space, it is straightforward to trace all its memory consumption, which involves the main process itself, plus its children processes, and all required dynamic libraries. In the process of extracting the XDP memory data, memory deltas have been made from `/proc/meminfo` which lists the memory occupied by the kernel stack, both dynamic and static. Therefore, a memory difference was obtained, taking samples before anchoring the bytecode, and once it was already loaded in the kernel. This procedure has been carried out in an iterative way, until achieving the data observed in Figure 6, which goes up to just 16KB.

It is interesting to note that, with XDP, we are only taking into account the amount of memory needed to load the bytecode in the Kernel, leaving out the amount of memory of the different kernel blocks which XDP requires to work with, which are common to other services, since they are already loaded in memory. Therefore, in order to have some equity between both measures in memory, in P4, only the memory that consumes the processes associated with the BMv2 was considered, taking out from the total the necessary memory of the dynamic libraries, common as well to other processes of user space.

### 4.2.3. CPU Consumption

As in the previous measurement, obtaining reliable data from XDP was a challenge because it operates in Kernel space, while in P4, BMv2 was leveraged to run the P4 code, being a user space process, it has been possible to collect all its CPU consumption without problems. The methodology employed has been similar to the latency study in Section 4.2.1, however, in this case, the SUT machine has been stressed much more by generating packets of 64 bytes with the pktg3n [40] tool at different rates from the TGR machine. The CPU information was obtained using the `mpstat` program with a minimum interval between readings so that the precision of the CPU usage was the maximum possible.

To obtain the CPU usage in XDP, the methodology described in one specific article that compared XDP and OVS (XDP vs. OVS: https://people.kernel.org/dsahern/xdp-vs-ovs) was applied. This methodology involves disabling the distribution of packet processing between the different cores of the machine, associating the queues of the interface where the packets are going to arrive to a single core and making this core work with a 100% user space process. Once completed, the CPU consumption per `%softirq` must be analyzed, which is the percentage of CPU destined to attend to software interruptions, in this case, to process the arriving packets with XDP.

It is important to point out that this CPU must work with a unique user process, since otherwise XDP will be stealing processing time equally from all the user space processes that are executed, making it impossible to quantify the associated `%softirq` for the processing of the incoming packets. In this test, following the methodology mentioned before, the `openssl speed` command has been used to get the CPU at 100%.

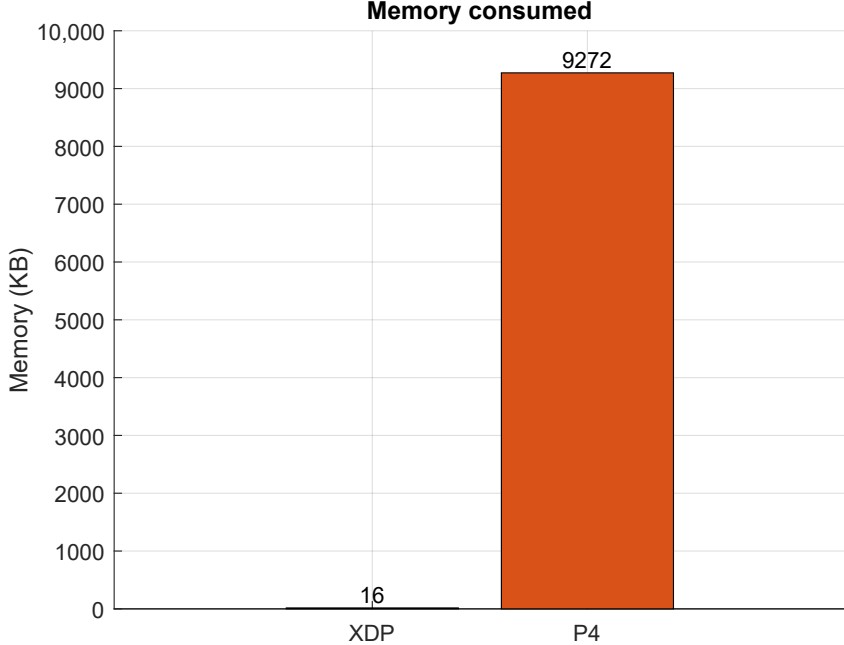

**Figure 6.** Memory consumption without shared objects of XDP vs. P4.

As indicated on the testbed, the SUT machine only had one core, so most of the procedures required to obtain the CPU usage are simplified. In this case, as the amount of packets increases, the `%softirq` will increase while the `%usr` will decrease, proving how the user space process is interrupted a greater number of times due to the fact that a greater number of packets have to be processed.

As illustrated in Figure 7 (Kpps = $10^3$ packets/s), the difference between the two technologies is quite significant, with XDP clearly in favour. These results reflect the clear difference in performance that involves processing the packets at the earliest point of the Kernel, compared to processing them with the BMv2, in user space. Therefore, there are current developments [41] that want to take advantage of this performance difference, by translating functionalities described in P4 to XDP.

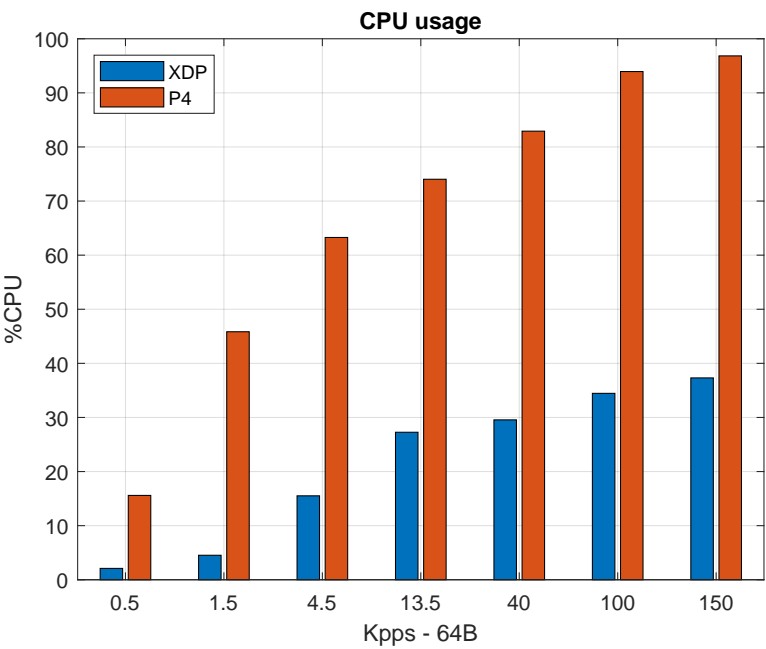

**Figure 7.** CPU consumption of XDP vs. P4.

### 4.3. Qualitative Review

The aim of this third and final analysis is to indicate that, regardless of the performance results previously obtained for XDP and P4, there are certain points of a qualitative nature that must be taken into account when evaluating which technology to use in the integration of IoT devices in SDN networks.

Looking at Figure 8, we can see the assessment of certain points of a qualitative nature that each of the technologies has. In the case of documentation, we evaluated how easy was to implement the use cases with the information available. Although in both cases, some details were missing, P4 was more complete. As for hardware and SDN integration, we examined current hardware vendors and standards, obtaining more support for P4, possibly because its research community is directly linked to the SDN field. Finally, migration and updates, and independence from other software, is more favourable for XDP as it just depends on Linux-based systems and has few dependencies. In summary, if we observe the strongest points of P4, these are the quality of the documentation generated for its users and the potential for immediate integration with SDN networks by having an interface already defined for the controllers. However, it has several dependencies that make it suffer greatly from migrations and version upgrades. Regarding XDP, its greatest strength is its independence from other software, as it only requires the device in question to carry the Linux Kernel, while its potential for integration with SDN networks is still very immature, as it does not have a defined control interface. In any case, this qualitative analysis is limited and subjective, hence a more comprehensive study is required to have a precise overview.

Additionally, and according to Figure 9, the trend, in terms of publications in both technologies, is increasing. Nevertheless, it can be observed that the number of P4 publications is much higher. Looking at the trend of repositories (`active/total`) for each technology, the repositories created in GitHub for both technologies reflect an upward trend, but once again, P4 is ahead of it. For all the above reasons, even taking into account the great difference in performance between the two technologies, sometimes qualitative factors make the difference in use.

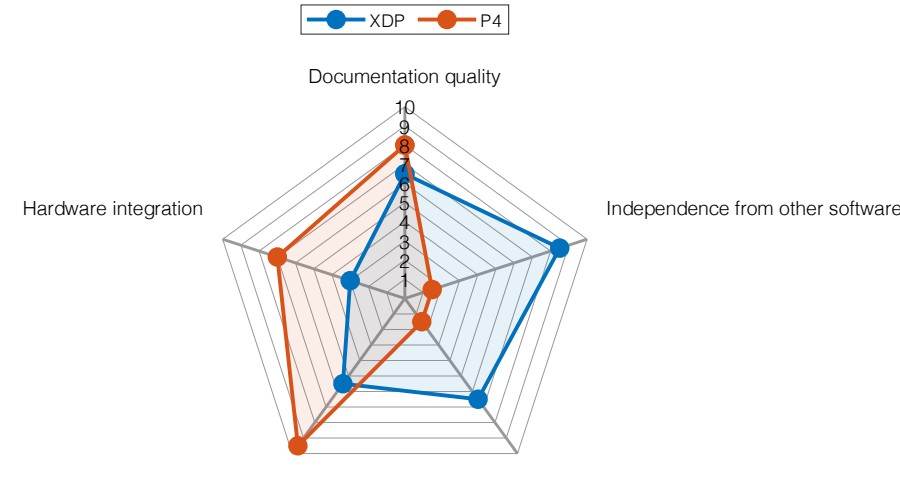

**Figure 8.** Qualitative analysis of XDP and P4 based on 5 parameters.

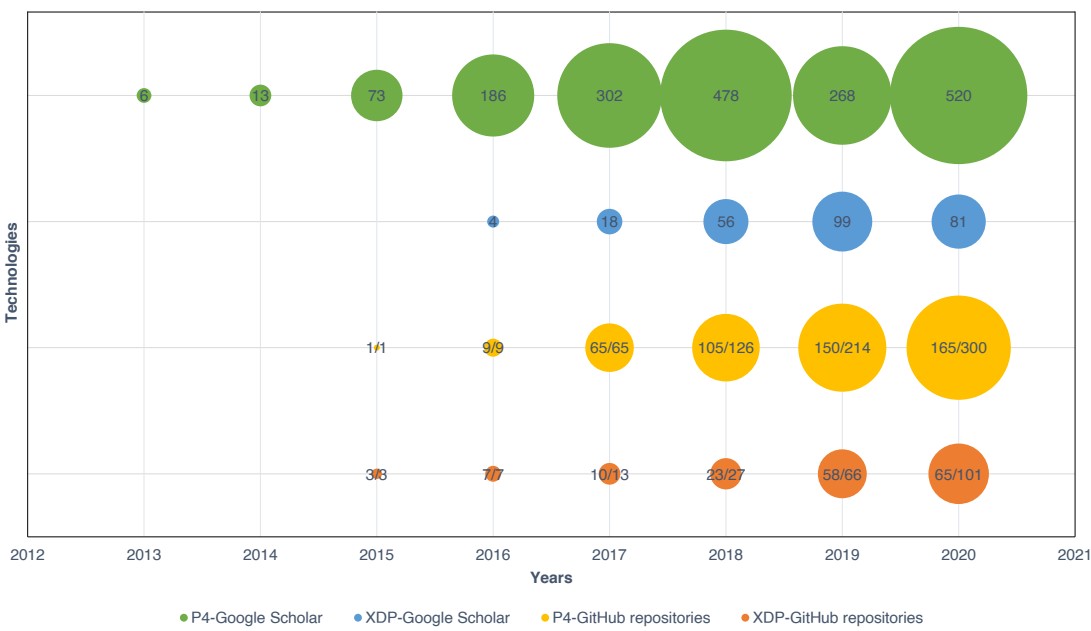

**Figure 9.** Trend analysis of XDP and P4 in Google Scholar and GitHub.

## 5. Discussion

Considering the evaluations performed for both XDP and P4, the first conclusion is that the simplest functionality cases (drop/pass/echo) are easier to implement in XDP, while actions requiring some type of packet processing/logic or packet creation/duplication are most suited for P4. This could make sense, as XDP could be considered a manager of interface traffic, while P4 is envisioned to handle packets similarly to traditional forwarding devices. Therefore, we could state that XDP is more advisable for simpler IoT devices, while P4 could be deployed in more complex devices, such as IoT gateways.

Furthermore, this initial conclusion is confirmed by the second analysis, testing performance in Linux-based systems. Nevertheless, it is important to note that all P4 tests required a target and, in our case, BMv2 was leveraged, which is still a prototype and fails to accomplish high-performance results. Therefore, a second conclusion could be that an enhanced and tailored version of BMv2 could be valuable to test IoT environments, particularly when no hardware targets are available (e.g., NetFPGA).

An idea to achieve this could be defining a minimum set of primitives that constrained IoT motes should implement to operate in LLNs, and then design the new target for P4 accordingly.

Additionally, the third analysis illustrate how there is a growing trend for both technologies and how they accomplish complementary objectives. Therefore, a final conclusion could be that there is no clear winner, but instead, XDP or P4 could perfectly work together [32] to match different requirements in IoT deployments.

Finally, just out of curiosity, if we pay attention to the number of code lines, as depicted in Figure 10, which are necessary to develop the functionalities described in Section 3.1, we can appreciate how P4 requires a homogeneous amount of lines, since there is always a need to implement the different blocks of the V1Switch architecture (https://github.com/p4lang/p4c/blob/master/p4include/v1model. p4). Whereas, XDP demands a minimal number of lines when the functionality implemented is simple, but as this functionality grows, its complexity and lines of code increase too [41]. Therefore, although with P4 it is required to implement different pre-set blocks in the V1Switch architecture, it makes the development of functionalities easier, since the effort and amount of work does not vary much depending on the complexity of the functionality to be developed.

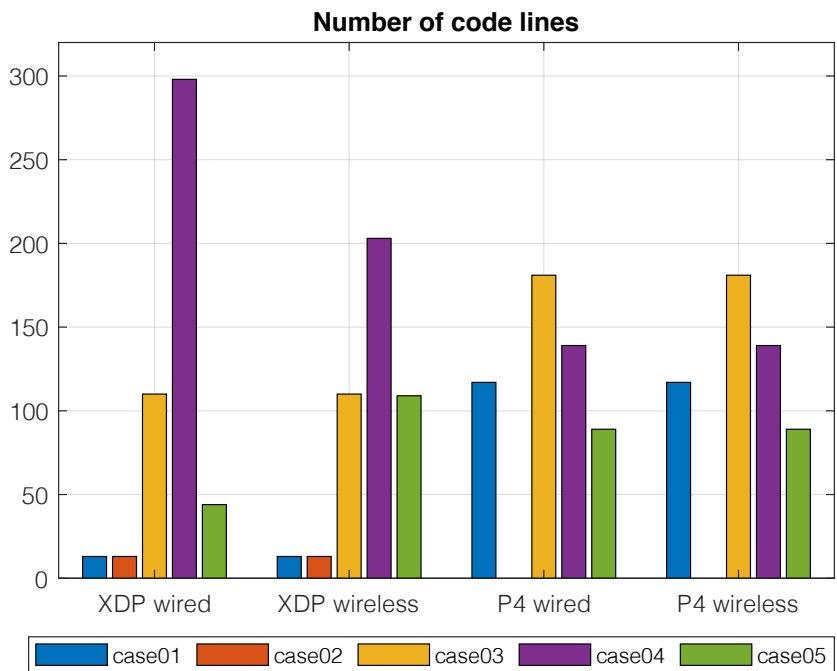

**Figure 10.** Number of code lines for each use case in both XDP and P4, and wired and wireless scenarios.

Moreover, we would like to point out the difference involved in debugging the functionalities developed with one technology or another. When we are working with the BMv2 target to test P4 code, we have different ways to debug, through log files, via console, Inter-Process Communication (IPC) mechanisms from which we can trace all the operations that have been carried out with a packet in the processing pipeline. On the other hand, there is XDP, which by working at such a low level, makes debugging a much more complicated and less accessible task for the average user.

In summary, XDP and P4 are complementary, and their use depends on the application of the device that makes use of them. Both are growing fast under the 5G networks paradigm and are promising technologies for the next IoT challenge under 6G networks. However, each one has its own features that make them ideal for different applications:

- XDP works well with simple message actions (drop, pass, and echo), which makes it ideal for devices with low traffic load and limited resources as final devices (motes) in the IoT paradigm. Some examples of these real-world scenarios encompass Industry 4.0 [42] or RPL-based networks [43] (monitoring of farming or sewage systems).

- P4 is oriented to more complex systems with high traffic load that require a high rate of processing packet, which makes it ideal for the devices in the core of the networks (i.e., switches). Additionally, they could also foster the implementation of fog and edge systems supporting IoT implemented with XDP, for example serving as collectors of data, applying AI [44] or providing high throughput at the edge (required by applications like video streaming, Augmented Reality (AR), Virtual Reality (VR), etc. [45]).

It is important to note that, despite the moderate performance of P4 shown in the evaluation, there are hardware specific platforms that drastically increase P4 performance, such as network-oriented FPGA devices (NetFPGAs), which combine the flexibility that P4 language provides with high-speed electronic boards to obtain high-performance devices. Thus, both technologies can coexist since they are complementary, each one occupying its market niche.

## 6. Conclusions and Future Works

In this article, we analyzed the suitability of P4 and XDP to provide enhanced programmability to IoT devices, a required feature for 6G networks and beyond. For this purpose, we performed several analyses, quantitative and qualitative, checking diverse use cases and performance among others. The main conclusion is that both technologies are recent and have a clear growing trend, and they will be a pillar for future 6G deployments. At the same time, while XDP is particularly appropriate for constrained IoT motes (that is LLNs), P4 is also a great choice for IoT gateways, for example.

For future works, we believe it would be very valuable to test both XDP and P4 in real constrained targets, such as those devoted to deploying LLNs. In particular, it could be interesting to implement the RPL using either XDP or P4. This would also imply the selection and design of a set of fundamental network primitives for these constrained environments. Moreover, the most common software P4 target so far, BMv2, severely limits the performance of P4. Thus, we anticipate P4 could be really promising for constrained IoT environments as well if a special effort was made to improve the current version of BMv2, to match the requirements for testing these scenarios.

The fast-growing rate of new-generation networks advocates for a fully programmable network infrastructure to incentivize a quick adaptation of all its devices to new network functionalities and services. XDP and P4 are two frameworks capable of achieving this requirement. Therefore, their use will increase in the next few years as they provide fast programming capabilities to update or enhance network devices with high independence of their underlying hardware.

**Author Contributions:** Conceptualization, E.R. and J.A.-H.; methodology, E.R. and J.A.-H.; software, D.C.; validation, D.C. and J.A.-H.; formal analysis, J.A.-H. and I.M.-Y.; investigation, E.R. and I.M.-Y.; resources, E.R. and I.M.-Y.; data curation, D.C. and D.L.-P.; writing–original draft preparation, all authors; writing–review and editing, E.R. and I.M.-Y.; visualization, D.L.-P.; supervision, E.R. and I.M.-Y.; project administration, E.R. and I.M.-Y.; funding acquisition, E.R. and I.M.-Y. All authors have read and agreed to the published version of the manuscript.

**Funding:** This research was funded by grants from Comunidad de Madrid through Project TAPIR-CM (S2018/TCS-4496) and Project IRIS-CM (CM/JIN/2019-039), and from Junta de Comunidades de Castilla la Mancha through Project IRIS-JCCM (SBPLY/19/180501/000324).

**Conflicts of Interest:** The authors declare no conflict of interest. The funders had no role in the design of the study; in the collection, analyses, or interpretation of data; in the writing of the manuscript, or in the decision to publish the results.

## Abbreviations

The following abbreviations are used in this manuscript:

| | |
|---|---|
| AI | Artificial Intelligence |
| API | Application Programming Interface |
| ARP | Address Resolution Protocol |
| BMv2 | Behavioral-Model |
| BPF | Berkeley Packet Filter |

| DPDK | Data Plane Development Kit |
|---|---|
| eBPF | extended Berkeley Packet Filter |
| HDDP | Hybrid Domain Discovery Protocol |
| IoT | Internet of Things |
| IPC | Inter-Process Communication |
| IRQ | Interrupt Request |
| ITU | International Telecommunication Union |
| LLN | Low-Power and Lossy Network |
| LOADng | Lightweight On-demand Ad hoc Distance-vector routing protocol-next generation |
| MEC | Multi-access Edge Computing |
| OVS | Open vSwitch |
| P4 | Programming Protocol-independent Packet Processors |
| SDN | Software-Defined Networking |
| SUT | System under Test |
| TC | Traffic control |
| TGR | Traffic Generator and Receiver |
| XDP | Express Data Path |

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
