# Peer review of "Analysis of P4 and XDP for IoT Programmability in 6G and Beyond"

_2624-831X, doi:10.3390/iot1020031_

Round 1

Reviewer 1 Report

The manuscript presents  an analisys of P4 and XDP applied to IoT. The article is well organized and the results are clearly presented. I have only one comment:

   - In [1], an analysis of P4 and XDP is performed. This article should be referenced in the manuscript, and the consistency of results presented in this reference and those carried out by authors should be checked and commented in Section 4.

References:

[1] Tu, William, Fabian Ruffy, and Mihai Budiu. "Linux Network Programming with P4." Linux Plumbers’ Conference 2018. 2018.

Reviewer 2 Report

The paper presents an analysis of P4 and XDP for IoT programmability.

The discussion section should include more details related to the relevance of the results to different real-world scenarios.

Reviewer 3 Report

The manuscript presents a comparison between two technologies, P4 and XDP, used to program Software-Defined Networking environments.

The authors have prepared a series of use cases to test the technologies under different aspects, and they have provided an online repository to allow for test replication.

The overall approach followed by the authors seems sound, and the claimed results are in line with the proposed experiments.

Reviewer 4 Report

The manuscript "Analysis of P4 and XDP for IoT programmability in 6G and beyond" analyzes the suitability of P4 and XDP to provide enhanced programmability for IoT devices in the light of 6G. The proposal is clear and the topic is interesting and timely. The results are convincing and main finding are relevant. The overall writing style is fair whereas a deep proof-reading is required (see points 3-7). Also the effectiveness of the paper in its main points should be enhanced (see points 1,9, 10). The presentation requires some interventions too (see points 2, 8) and I suggest using Latex for a better look-and-feel. Detailed comments:
1: move the oultine of paper's contributions from section 2 to introduction. Then, insert a table or simply discuss the relationships (in terms of similarities, differences, innovation) of the proposal with respect to the related works.
2: double-check the caption style of figures and tables which looks incongruent (e.g., in Fig. 2 the caption is below the figure, in Fig 3 the caption is above it; the same holds for tables 1 and 2)
3:Did you mean "working"? Or maybe you should add a pronoun? In active voice, 'require' + 'to' takes an object, usually a pronoun.
4:interface.In -> interface. In
5:a final conclusions
6:funcionality->functionality
7:several analysis-> several analyses
8:reduce blank space between Fig. 8 and its caption.
9: highlight main findings of section 5 thorugh a summarizing bullet point at the end of the section
10: outline future works within the conclusion
11: improve the bibliography: cite <Taivalsaari, Antero, and Tommi Mikkonen. "A roadmap to the programmable world: software challenges in the IoT era." IEEE Software 34.1 (2017): 72-80> and <Omnes, Nathalie, et al. "A programmable and virtualized network & IT infrastructure for the internet of things: How can NFV & SDN help for facing the upcoming challenges." 2015 18th International Conference on Intelligence in Next Generation Networks. IEEE, 2015> as general reference for SDN/IoT programmability; cite <Fortino, Giancarlo, et al. "Modeling Opportunistic IoT Services in Open IoT Ecosystems." WOA. 2017> beside [4]; cite <Sabella, Dario, et al. "Mobile-edge computing architecture: The role of MEC in the Internet of Things." IEEE Consumer Electronics Magazine 5.4 (2016): 84-91> besides [14]; edit [5,40,41] as footnotes.

In conclusion, the paper has merit but a minor revision is required to fix the issues reported above.

Round 2

Reviewer 1 Report

Authors have addressed my comments and suggestions.